# Modulation of EZH2 Activity Induces an Antitumoral Effect and Cell Redifferentiation in Anaplastic Thyroid Cancer

**DOI:** 10.3390/ijms24097872

**Published:** 2023-04-26

**Authors:** Diego Claro de Mello, Kelly Cristina Saito, Marcella Maringolo Cristovão, Edna Teruko Kimura, Cesar Seigi Fuziwara

**Affiliations:** Department of Cell and Developmental Biology, Institute of Biomedical Sciences, University of Sao Paulo, Sao Paulo 05508-000, Brazil

**Keywords:** EZH2, anaplastic thyroid cancer, CRISPR/Cas9

## Abstract

Anaplastic thyroid cancer (ATC) is a rare and lethal form of thyroid cancer that requires urgent investigation of new molecular targets involved in its aggressive biology. In this context, the overactivation of Polycomb Repressive Complex 2/EZH2, which induces chromatin compaction, is frequently observed in aggressive solid tumors, making the EZH2 methyltransferase a potential target for treatment. However, the deregulation of chromatin accessibility is yet not fully investigated in thyroid cancer. In this study, EZH2 expression was modulated by CRISPR/Cas9-mediated gene editing and pharmacologically inhibited with EZH2 inhibitor EPZ6438 alone or in combination with the MAPK inhibitor U0126. The results showed that CRISPR/Cas9-induced *EZH2* gene editing reduced cell growth, migration and invasion in vitro and resulted in a 90% reduction in tumor growth when EZH2-edited cells were injected into an immunocompromised mouse model. Immunohistochemistry analysis of the tumors revealed reduced tumor cell proliferation and less recruitment of cancer-associated fibroblasts in the EZH2-edited tumors compared to the control tumors. Moreover, EZH2 inhibition induced thyroid-differentiation genes’ expression and mesenchymal-to-epithelial transition (MET) in ATC cells. Thus, this study shows that targeting EZH2 could be a promising neoadjuvant treatment for ATC, as it promotes antitumoral effects in vitro and in vivo and induces cell differentiation.

## 1. Introduction

Anaplastic thyroid cancer (ATC) is the rarest form of thyroid cancer (1–2% of cases) [1], and as an undifferentiated cancer, it is refractory to classical chemotherapy and radioiodine therapy [2]. The genes *NIS*, *TPO* and *TG* play crucial roles in thyroid differentiation, as they are required for iodine trapping (NIS), metabolism and hormone synthesis (TPO and TG). These genes are regulated by thyroid-specific transcription factors PAX8, NKX2-1 and FOXE1, which are often dysregulated in ATC patients [3]. In fact, thyroid cell dedifferentiation is associated with cancer progression and aggressiveness as ATC tumor cells show no uptake of radioiodine and become resistant to radioactive iodine therapy [4,5,6]. 

Besides dedifferentiation, some ATC features include rapid cell growth and invasion of adjacent tissues, which results in high morbidity and mortality of patients within a few months [7,8]. The molecular pathways altered in ATC pathogenesis are associated with driver mutations in MAPK signaling genes, such as *BRAF^T1799A^* and *RAS*, *TP53* and *TERT* promoter mutations [9,10]. *TERT* promoter mutations (C228T and C250T) increase transcription of the *TERT* gene [11], while *TP53* mutations lead to inactivation of the p53 protein [12]. These mutations are highly prevalent in ATC patients harboring MAPK mutations and are associated with poorer clinical outcomes [13,14,15,16]. Despite some progress in targeted therapy and our current understanding of ATC oncogenesis, curative therapy for ATC remains a challenge for oncologists [17]. Therefore, there is an urgent need to identify new target molecules involved in ATC biology.

Recently, epigenetic reprogramming has emerged as a novel cancer hallmark [18]. The Polycomb-group proteins are known epigenetic modulators of cell proliferation, differentiation and organ development [19]. The Polycomb Repressive Complex 2 (PRC2) plays a crucial role in chromatin and gene expression repression, primarily by binding to the promoter regions of target genes and catalyzing the trimethylation of H3 histone on lysine 27 (H3K27me3) [20,21]. PRC2 proteins are active during early developmental stages but typically silenced in adult tissues. However, in aggressive solid tumors, PRC2 proteins can be reactivated, resulting in aberrant gene expression patterns that promote tumor growth and metastasis [19,22]. As the core catalytic subunit of PRC2, EZH2 plays a critical role in mediating its methyltransferase activity, which is essential for the deposition of H3K27me3 on target genes. However, to achieve this function, EZH2 requires the assistance of other PRC2 components, such as EED and SUZ12 [23,24].

The deregulation of PRC2 activity has been implicated in the development and progression of various types of cancer [25,26,27,28,29] and represents a potential target for the development of new cancer therapies. For instance, Tazemetostat (EPZ6438), an FDA-approved inhibitor of EZH2 methyltransferase, has been shown to be effective in treating epithelioid sarcoma and follicular lymphoma [30,31]. In thyroid cancer, EZH2 overexpression has been reported in ATC [32,33,34], and RNA interference modulation resulted in a reduction in cell migration and invasion in vitro in the 8505C cell line [32]. Furthermore, in papillary thyroid cancer (PTC) cells, a recent study has shown that combined treatment with EPZ6438 and dabrafenib or selumetinib enhances the differentiation of PTC cells [35]. Nevertheless, further investigation is needed to determine the role of PRC2/EZH2 in thyroid cancer aggressiveness and dedifferentiation by implementing the use of gene-editing techniques, such as CRISPR/Cas9 in the ATC model. 

Here, we demonstrate that targeting EZH2 using CRISPR/Cas9 or EPZ6438 results in significant antitumor effects in ATC cells, including reduced cell proliferation, colony formation, invasion and migration in vitro, as well as reduced tumor growth in vivo. Additionally, EZH2 inhibition led to enhanced differentiation of thyroid follicular cells and induced a mesenchymal–epithelial transition (MET). These findings suggest that targeting EZH2 may represent a promising therapeutic strategy for ATC by promoting thyroid cell differentiation and restoring the epithelial phenotype.

## 2. Results

### 2.1. EZH2 Is Overexpressed in Thyroid Cancer

To determine whether EZH2/PRC2 components are deregulated in PTC and ATC, we first analyzed the expression of EZH2, EED and *SUZ12* in BRAF^V600E^ thyroid cancer cell lines (Figure 1A,B). Consistent with previous reports (Borbone, Troncone et al. 2011), EZH2 expression was upregulated specifically in ATC cell lines, whereas EED was upregulated only in ATC cell line SW1736. No difference in *SUZ12* mRNA levels was observed between cells (Figure 1A). We also accessed data from public microarray and RNAseq databases and confirmed that EZH2/PRC2 genes are overexpressed in ATC patients, compared to nontumoral matched samples in TCGA and GEO databases. Interestingly, in PTC, we already detected an increase in *EZH2* and *SUZ12* mRNA levels compared to nontumoral matched samples in TCGA (Figure 1C,D).

### 2.2. CRISPR/Cas9-Mediated EZH2 Gene Editing Improves Differentiation of ATC Cells

Next, we targeted the *EZH2* gene with the CRISPR/Cas9 gene-editing tool in ATC cell lines SW1736 and KTC2 (Figure 2 and Appendix A) to modulate EZH2 expression. *EZH2* is a 20-exon gene located at chromosome 7 and gene editing was performed by guiding Cas9 to the second exon, which contains the start codon (*ATG*) using different sgRNAs (sg7, sg18 and sg25) (Figure 2A).

Therefore, we first analyzed EZH2 protein levels in mixed populations of SW1736- and KTC2-edited cells (Figure 2B and Appendix A). Although the three sgRNAs reduced EZH2 protein levels, the sg18 was the most efficient in SW1736 and KTC2. Then, we derived stable clones of SW1736-sg18 (ClA, ClC and ClE) and KTC2-sg18 (Cl3, Cl4, Cl5 and Cl9) after limiting dilution. The clones derived from SW1736-sg18 and KTC2-sg18 maintained reduced EZH2 expression (~50–70%) by generating truncated proteins (Figure 2C and Appendix A). Since we did not achieve a complete gene knockout using CRISPR/Cas9, we wonder if EZH2 methyltransferase activity diminished in SW1736-sg18 clones. Then, we assessed H3K27me3 levels and total H3 by Western blot in SW1736-sg18 clones and observed that SW1736-ClA and SW1736-ClE showed a dramatic reduction in H3K27me3 levels (Figure 2C). In SW1736-ClC cells, however, H3K27me3 increased instead, indicating that even with reduced EZH2 expression, its activity remained.

We confirmed that CRISPR/Cas9 + sgRNA18 specifically targeted the *EZH2* gene by sequencing genomic DNA from all clones (Figure 2D and Appendix A). The analysis using the SeqScreener Gene Edit Confirmation App showed that *EZH2*-mediated gene editing was successful with an overall efficiency of ~92–98% in SW1736 (Appendix A). Interestingly, the analysis of PCR amplification of the sg18-targeted region revealed two fragments of the *EZH2* gene in agarose gel electrophoresis for SW1736-clones genomic DNA (Appendix A). The DNA sequencing analysis of both fragments showed heterogeneity in gene editing (Figure 2D and Appendix A), that could be related to the SW1736 cells’ polyploidy [36], which increased the difficulty to target all additional copies of *EZH2.* For instance, 11.91% and 7.28% of SW1736-ClA sequences show an 18 nt and 22 nt deletion, respectively (Appendix A). Similarly, 30.39% and 20.26% of SW1736-ClC sequences showed a 9 nt and 11 nt deletion, respectively (Appendix A). In SW1736-ClE, the top indel sequences showed deletions of 14 nt (17.22%), 10 nt (14.76%), 4 nt (14.12%) and 7 nt (14.04%) (Appendix A). Finally, we also validated the off-targeting potential of sg18 through DNA sequencing and observed that sg18 did not edit *TMEM74B* and *MAD1L1*, two potential off-targets at coding genes in SW1736 clones (Appendix A). For KTC2 cell lines, we observed similar results with an overall efficiency of ~97–100% (Appendix A), and in the clones KTC2-Cl4 and Cl5, ~85% of sequence traces showed a 1 nt addition, a 9 nt or a 14 nt deletion, respectively (Appendix A).

Next, we performed an analysis of available ChiP-seq (chromatin immunoprecipitation sequencing) data in the Cistrome database and observed that EZH2 and H3K27me3 are specifically or mutually enriched at CpG-rich islands in promoter regions of thyroid-differentiation genes in non-thyroid cells (Appendix A). Then, we investigated the effects of CRISPR/Cas9-induced *EZH2* gene editing in ATC thyroid-cell differentiation. We observed an overall increased expression of iodine metabolism genes *NIS*, *TG*, *TSHR* and *GLIS3*, with a stronger effect in *TSHR* and *NIS* in SW1736 clones. However, surprisingly, we observed the downregulation of transcription factors *NKX2-1*, *FOXE1* and PAX8 levels in SW1736 clones (Figure 3A,B). Next, we asked if SW1736-ClE cells, which showed decreased H3K27me3 deposition (Figure 2C) and increased *TSHR* mRNA, would respond to exogenous TSH treatment, improving thyroid differentiation of ATC cells. We observed no significant improvement in thyroid-cell differentiation in SW1736-ClE or SW1736-CTR in response to treatment with 3 mU/mL TSH. The only exceptions were *TSHR* and *GLIS3* mRNA, which increased in both cells after TSH treatment (Figure 3C).

### 2.3. CRISPR/Cas9-Mediated EZH2 Gene Editing Induces Mesenchymal–Epithelial Transition (MET)

Epithelial–mesenchymal transition (EMT) is a term often used to describe how polarized cells undergo complex molecular changes to shift to a mesenchymal-like phenotype. EMT, in fact, improves metastatic potential and is frequently observed in aggressive tumors [37]. Therefore, we analyzed whether *EZH2*-mediated gene editing with CRISPR/Cas9 modulated EMT.

In cell culture, we observed a marked epithelial-like phenotype in SW1736 clones, particularly SW1736-ClC and ClE, with cells juxtaposed in colonies (Figure 4A). SW1736 is derived from ATC which is known to exhibit high levels of EMT transcription factor ZEB1 and low levels of E-cadherin (Figure 4B,C). The morphological changes in SW1736 clones reflect a reversal of EMT, a process known as MET (mesenchymal–epithelial transition). Indeed, we observed a significant increase in the expression of E-cadherin at both mRNA and protein levels in the SW1736-edited cells, while ZEB1, ZEB2 and N-cadherin were downregulated (Figure 4B,C).

The EMT process is regulated by a feedback loop between ZEB1 and *miR-200* family to regulate E-cadherin levels [38]. We observed that *EZH2*-mediated gene editing with CRISPR/Cas9 led to the induction of *miR-200a* in the epithelial-like SW1736-ClC and SW1736-ClE cells. In addition, SW1736-ClC showed an upregulation of mature *miR-200c* (Figure 4D,E). It is interesting to note that the *miR-200a* basal levels were more than 150-fold higher than those of *miR-200c* which may influence the EMT/MET process. 

The Wnt/β-catenin signaling pathway is known to be overactivated in ATC [39,40,41,42], leading to nuclear localization of β-catenin and its binding to TCF/LEF, ultimately inducing *MYC* and *ZEB1* transcription while repressing *miR-200a* transcription [43,44]. To assess Wnt signaling activation, we used the M50 luciferase reporter plasmid, which contains seven sites for TCF/LEF factors. We observed reduced TCF/LEF luciferase activity in SW1736-ClC and ClE, indicating Wnt signaling repression (Figure 4G). Western blot analysis revealed that total β-catenin was repressed in SW1736-ClA and ClE (Figure 4F), and downstream Wnt signaling targets, such as ZEB1 and c-MYC, were downregulated in SW1736 clones (Figure 4B,C,F). Interestingly, although total β-catenin was not reduced in SW1736-ClC, Wnt signaling was still inhibited. This may be due to other predicted targets for *miR-200a*, such as *AXIN1* and *TCF4*, and for *miR-200c*, including *WNT1* and *TCF4* that could lead to Wnt signaling repression. Collectively the results suggest that *EZH2*-mediated gene editing using CRISPR/Cas9 has the potential to reverse the EMT process in ATC and may have the participation of *the miR-200* family and the inhibition of Wnt signaling.

According to the Cistrome database, EZH2 or H3K27me3 appears to have distinct roles in regulating gene expression and EMT phenotype in different types of cells. In epithelial-like cells, EZH2 and H3K27me3 seem to be enriched at promoter regions of mesenchymal genes, while in mesenchymal-like cells, EZH2 appears to be enriched at promoter regions of epithelial genes (Appendix A). For example, in induced pluripotent stem cells (iPSC) and prostate cancer cells, mesenchymal genes, such as *ZEB1*, *ZEB2*, *CDH2*, *TWIST* and *SNAI1/2*, are enriched with EZH2 binding, but not epithelial genes such as *CDH1* or *SNAI3*. In contrast, fibroblasts, embryonic stem cells (ESC) and melanoma cells show enrichment of EZH2 or H3K27me3 at epithelial genes’ promoter regions but not at the mesenchymal ones. Thus, these findings suggest that EZH2 and H3K27me3 play different roles in regulating EMT, which could have implications for understanding the molecular mechanisms underlying these cellular states in different cell types.

### 2.4. CRISPR/Cas9-Mediated EZH2 Gene Editing Has an Antitumoral Effect In Vitro 

We evaluated the impact of EZH2 loss of function on cell migration, invasion and cell counting assays in vitro. Our results demonstrate that SW1736-ClC and SW1736-ClE cells exhibit a ~6-fold reduction in cell migration and ~5-fold reduction in cell invasion compared to control cells (Figure 5A,B). Additionally, SW1736-edited clones exhibit a significant decrease in cell count after 72 h of culture (Figure 5C). Colony formation assay revealed that SW1736-ClA and SW1736-ClE form significantly fewer colonies than SW1736-CTR, while SW1736-ClC shows the opposite effect (Figure 5D). To investigate if the reduction in Wnt/β-catenin signaling (Figure 4F,G) mediated the reduced cell growth in SW1736-edited cells, we performed a β-catenin rescue assay. We transiently transfected β-catenin into SW1736-CTR and SW1736-edited cells and evaluated the colony formation ability. Interestingly, we noted that overexpression of β-catenin rescues/enhances colony formation in SW1736-edited cells, but this effect was not observed in SW1736-CTR-unedited cells (Appendix A). This finding suggests that the inhibition of the Wnt/β-catenin pathway may play a role in the antitumor effect observed in SW1736-edited cells. Altogether, these results show that modulation of EZH2 expression with CRISPR/Cas9 has an antitumoral effect in vitro.

### 2.5. CRISPR/Cas9-Mediated EZH2 Gene Editing Has an Antitumoral Effect In Vivo

In order to investigate tumor formation of CRISPR/Cas9-edited cells in vivo, we used the xenograft nude mouse model. We observed that SW1736-ClE exhibited a significant reduction in the volume of approximately 90%, as compared to SW1736-CTR tumors after 35 days of injection (Figure 6A,B). Histological analysis of tumor sections using H&E staining revealed that tumors from SW1736-control group displayed a solid pattern of epithelioid pleomorphic cells with high cellularity and intermingled with desmoplastic stroma, a typical characteristic of ATC (Figure 6C). In contrast, tumors from the SW1736-ClE group had lower cellularity of epithelioid cells but instead had large areas of dense connective tissue (Figure 6C).

The immunohistochemical analysis revealed that SW1736-CTR tumors exhibited increased α-smooth muscle actin (α-SMA) staining, a marker for cancer-associated fibroblasts (CAFs) [45] that are typically found in the desmoplastic stroma of ATC (Figure 6D,E). In addition, consistent with the findings regarding tumor volume and cell counting data, SW1736-CTR tumors showed a high number of Ki67-positive cells, a marker for cell proliferation, in contrast to SW1736-ClE tumors (Figure 6D,E). Our results indicate that *EZH2*-mediated gene editing with CRISPR/Cas9 promotes an antitumoral effect in vivo.

### 2.6. Pharmacological Blockage of EZH2 Methyltransferase Activity Induces MET and Differentiation of ATC Cells and Reduces Colony Formation

To validate the results of CRISPR/Cas9-induced *EZH2* gene editing, we treated ATC cell lines SW1736 and KTC2 with EPZ6438, a pharmacological inhibitor of EZH2 that blocks its methyltransferase activity, for 6 days alone or in combination with MEK1/2 inhibitor U0126. Both KTC2 and SW1736 cell lines harbor BRAF^V600E^ mutation that constitutively activates MAPK signaling and is linked to cell proliferation and dedifferentiation. EPZ6438 treatment alone induced the expression of thyroid-differentiation genes, including *NIS*, *TPO*, *TG*, *TSHR*, *NKX2-1* and *GLIS3* (Figure 7A), but reduced the EMT transcription factor *ZEB1* and induced *CDH1* in KTC2 cells (Figure 7B). Combined treatment with U0126 resulted in an additive effect on *TSHR*, *NKX2-1*, *GLIS3* and *CDH1* genes. However, surprisingly, the induction of *NIS* and *TPO* was attenuated by the U0126 treatment (Figure 7A,B). In SW1736 cells, a similar effect was observed except for the higher basal expression of *PAX8*, *FOXE1* and *CDH1* genes than in KTC2 (Figure 7A,B). Furthermore, we observed that treatment with EPZ6438 reduced the colony formation ability in SW1736 and KTC2. Additionally, treatment with U0126 alone reduced colony formation in KTC2, but not in SW1736 cells, and no additional effect was observed in the combined treatment for both cells (Figure 7C). Our results indicate that the pharmacological inhibition of EZH2 activity induces redifferentiation and reduces colony formation ability.

## 3. Discussion

The overexpression of EZH2 is consistent in various types of solid tumors and associated with poor prognosis in prostate, breast, bladder, lung and pituitary cancers [25,26,27,28,29]. EZH2 is the catalytic domain of PRC2, but EED and SUZ12 are necessary for H3K27me3 deposition [21,46]. PRC2 also requires other noncore subunits to interact with the CpG-rich islands and histones, as well as to recruit and guide the complex to target regions, including JARID2, AEBP2, the chaperone RbAp46 and ncRNAs [47,48,49,50,51]. In thyroid cancer, overexpression of EZH2 is linked to aggressive behavior and dedifferentiation in ATC and poorly differentiated thyroid cancer patients [32,33,34]. Our findings support the observed high levels of EZH2 in ATC and reveal an increase in SUZ12 and EED expression. Additionally, we report a progressive increase in EZH2/PRC2 activity in PTC, as evidenced by increased *EZH2* and *SUZ12* expression in tumor tissue compared to adjacent nontumoral tissue in the TCGA database. The exclusive activation of EED suggests a potential role for this gene in the transition from nonaggressive to aggressive cancer.

Polycomb group proteins play a crucial role in normal cell and organ development. For instance, repression of the catalytic domain of PRC2, EZH2, directs the differentiation of human embryonic stem cells into mesodermal cells through the reduction in the global level of H3K27me3 [52]. Moreover, EZH2 has been shown to regulate pluripotency during myogenesis and epidermis differentiation, as its inhibition is observed in these processes [53,54]. Besides the canonical role of EZH2 in PRC2 that ultimately leads to H3K27me3 deposition, EZH2 may noncanonically interact with several transcription factors and chromatin modifiers, leading to the activation or repression of target genes in a PRC2-independent manner [45,55]. In cancer, noncanonically EZH2 has been reported to promote tumorigenesis through the activation of oncogenes and inhibition of tumor-suppressor genes in prostate cancer and breast cancer [56,57]. Therefore, dysregulation of PRC2/EZH2 activity may influence cell differentiation and tumorigenesis

In this study, we used both CRISPR/Cas9-mediated gene editing and pharmacological inhibition with the EPZ6438 inhibitor, alone or in combination with the MAPK inhibitor U0126, to achieve a permanent loss of EZH2 function and to temporarily inhibit its methyltransferase activity in the ATC cell lines SW1736 and KTC2. For gene editing, we used the ChopChop software to design three sgRNAs targeting the EZH2’s exon and chose the most efficient sgRNA (sg18) to generate edited clones. Despite observing only partial EZH2 knockout, *EZH2*-mediated gene editing was sufficient to reduce H3K27me3 activity in SW1736-ClA and SW1736-ClE cells. The effectiveness of CRISPR/Cas9-mediated gene editing may be influenced by factors such as gene copy number variation, in addition to sgRNA + Cas9 efficiency and input [58]. Indeed, SW1736 cells are tetraploid and have amplification in the 7q arm of chromosome 7 where *EZH2* is located, indicating the possibility of additional copies of the *EZH2* gene [36].

The symporter NIS (also known as SLC5A5) is a crucial thyroid-cell-differentiation gene responsible for iodine uptake into follicular cells to produce thyroid hormones [59]. However, in undifferentiated thyroid tumors such as ATC, cancer cells may progressively lose NIS expression and become unresponsive to radioiodine [9,60]. Our study demonstrated that modulating EZH2 activity with CRISPR/Cas9 or EPZ6438 increased thyroid follicular-cell differentiation by regulating the expression of iodine metabolism genes and thyroid transcription factors. Specifically, *EZH2* gene editing using CRISPR/Cas9 induced the expression of *NIS*, *GLIS3* and *TSHR* genes in ATC SW1736-edited cells. GLIS3 acts downstream of TSH/TSHR-mediated thyroid hormone biosynthesis as a transcriptional factor for *NIS* and *SLC26A4* (pendrin) transcription [61]. While TSHR expression increased in SW1736-edited cells, TSH treatment alone was insufficient to reactivate thyroid-differentiation genes in ATC upon EZH2 modulation with CRISPR/Cas9. When blocking H3K27me3 deposition with EPZ6438, we observed a significant enhancement in the expression of thyroid-differentiation genes, potentially through the upregulation of the NKX2-1 transcription factor. By the combined treatment of EPZ6438 with the MEK1/2 inhibitor U0126, this effect was only partially improved. Our findings are in part consistent with previous research that showed a combination of the EZH2 inhibitor EPZ6438 with the MAPK inhibitor selumetinib and TSH could increase the expression of thyroid follicular-cell differentiation genes and improve radioiodine uptake in PTC cells harboring BRAF^V600E^ [35]. These results indicate that EZH2 regulates thyroid-cell differentiation, but the synergy with other signaling pathways may influence the redifferentiation effect.

The deposition of H3K27me3 is a crucial step in chromatin remodeling, and it is highly enriched at CpG islands overlapping with PRC2-binding sites, which helps maintain transcriptional repression [20,62,63]. The Cistrome database [64] analysis reveals that EZH2 and H3K27me3 are strongly enriched at the CpG island of thyroid-differentiation genes NIS (*SLC5A5*) and *TSHR* and at the thyroid transcription factors *PAX8, NKX2-1* and *FOXE1*. Interestingly, EZH2 modulation with CRISPR/Cas9 and EPZ6438 treatment results in increased expression of *NIS* and *TSHR* in ATC cell lines, but PAX8 and NKX2-1, critical transcription factors involved in thyroid-differentiation-specific gene transcription [65,66,67], were downregulated in SW1736-edited cells. Indeed, it is important to note that SW1736 cells show high levels of *PAX8* [68], *FOXE1* and *CDH1*, (Figure 7A,B) compared to KTC2, indicating ATC tumor heterogeneity that may influence the response to EZH2 modulation.

In KTC2 and SW1736 ATC cell lines, treatment with the EZH2 inhibitor EPZ6438 resulted in redifferentiation through *NKX2-1*, whereas the MEK1/2 inhibitor primarily increased *PAX8* expression. Notably, EPZ6438 treatment had a more profound effect on thyroid differentiation by inducing the expression of *NKX2-1*. While previous reports have shown that EZH2 binds to the *PAX8* promoter in 8505C ATC cell line, resulting in transcriptional silencing that may impair ATC differentiation [32], our data demonstrate that EZH2 inhibition can lead to redifferentiation through the induction of *NKX2-1* expression. Supporting this finding, previous studies have shown that there are more predicted sites for NKX2-1 binding at the promoter regions of *Slc5a5*, *Tg*, *Tpo* and *Tshr* than for PAX8 [69]. Hence, collectively, this analysis highlights the importance of transcription factor binding in the regulation of thyroid-specific gene expression and the potential for EZH2 inhibition as a therapeutic strategy for ATC redifferentiation.

In this study, we also observed a mesenchymal–epithelial (MET) transition in EZH2-edited cells. This was characterized by increased E-cadherin levels and *miR-200a* expression and decreased ZEB1 and N-cadherin levels, which is consistent with a more epithelial morphology in vitro. Previous studies have shown that EZH2 plays a key role in the regulation of E-cadherin (*CDH1*) and *miR-200* expression [70,71,72] and that there is a regulatory loop of *miR-200* over components of PRC1 complex, including RING2 and BMI1, which are essential for chromatin condensation following H3K27Me3 deposition by EZH2. Indeed, we found that editing the EZH2 gene in SW1736-ClC and SW1736-ClE resulted in a significant increase in *miR-200a* expression. This increase in *miR-200a* may explain the observed downregulation of ZEB1 and promotion of MET, which was only partially achieved in SW1736-ClA. Furthermore, treatment of KTC2 cells with EZH2 inhibitor EPZ6438 induced downregulation of EMT genes and improved *CDH1* expression. In SW1736 cells, which have a higher basal level of *CDH1* than KTC2 cells, EPZ6438 did not modulate the expression of *CDH1* or *ZEB1*. Moreover, MEK1/2 inhibitor treatment induced modulation of EMT genes and improved *CDH1* expression in KTC2 cells, but not in SW1736 cells. Our data suggest that EZH2 regulates EMT in ATC cells but is influenced by cellular context.

Molecularly, it was shown that EZH2 silences Wnt antagonists, such as DKK1 and AXIN2, leading to Wnt/β-catenin signaling activation in hepatocellular carcinoma [73]. β-catenin, as a result of Wnt/β-catenin signaling activation, accumulates in the cytoplasm and translocates into the nucleus where it binds to LEF/TCF factors to induce transcription of its downstream effectors, such as *MYC* and *ZEB1* [43]. We found that *EZH2* gene editing in SW1736 cells results in a lower activity of Wnt/β-catenin signaling, which subsequently led to the inhibition of c-MYC and ZEB1 expression. The antiproliferative effect of *EZH2* gene editing may be explained by the reduction in c-MYC expression. c-MYC is a proto-oncogene that plays a crucial role in promoting cell growth and proliferation [74,75]. Accumulating evidence suggests that EZH2 functions as a transcriptional factor or coactivator in cancer, independent of PRC2, promoting oncogenesis through multiple pathways [55]. For example, EZH2 has been reported to interact noncanonically with c-MYC to promote oncogenesis [76]. Additionally, in ER^+^ breast cancer cells, EZH2 has been found to noncanonically interact with estrogen receptor α (ERα) and β-catenin, forming a ternary complex that binds to *MYC* promoter at the LEF/TCF-binding sites, thereby inducing its transcription [77]. Therefore, *EZH2* gene editing has the potential to disrupt these interactions, resulting in decreased expression of c-MYC and subsequent suppression of cell proliferation. On the other hand, ZEB1 is a transcription factor responsible for inducing a mesenchymal-like morphology by inhibiting *CDH1* and *miR-200a* transcription [38]. Previous studies have shown that low levels of *miR-200a* are associated with β-catenin overexpression, a *miR-200a* target, leading to the activation of Wnt/β-catenin signaling [78,79]. Our findings suggest a regulatory loop between EZH2, Wnt signaling and *miR-200* to control EMT in ATC.

Functionally, our findings demonstrate that modulation of the EZH2 gene using CRISPR/Cas9 methodology resulted in an antitumoral effect both in vitro and in vivo. The MET process inhibited cell migration and invasion in SW1736-ClC and SW1736-ClE. After 72 h, cell counting was greatly reduced in all SW1736-edited cells, while colony formation ability was reduced in SW1736-ClA and ClE, in contrast to SW1736-ClC. Indeed, the EMT-MET switch is a dynamic and complex process in which cancer cells shift through these phenotypes to favor metastasis or cell growth in colonies, respectively [43,80]. To this extent, we observed that part of the effect is mediated by β-catenin signaling as the rescue assay showed increased colony formation. Furthermore, EPZ6438 treatment in KTC2 and SW1736 reduced the colony formation, but no additional effect was observed when treating with MEK1/2 inhibitor. In our in vivo model, EZH2 modulation using CRISPR/Ca9 greatly reduced tumor growth and Ki-67-positive cells and exhibited a significant reduction in the number of cancer-associated fibroblasts (CAFs) in tumors derived from SW1736-edited cells compared to control. Notably, our previous report demonstrated that loss of *miR-200* induced Notch signaling in cancer cells, leading to the recruitment of CAFs and reshaping of the tumor microenvironment to favor metastasis and tumor growth in a lung adenocarcinoma mouse model [81]. Our functional results indicate the role of EZH2 in ATC tumor cell proliferation, invasion and migration.

We summarized the main findings in Figure 8.

## 4. Materials and Methods

### 4.1. Cell Culture and Treatments

#### 4.1.1. Cell Culture

The main cell line used was SW1736 derived from human anaplastic thyroid cancer. Other human thyroid cancer cell lines were used for gene expression and Western blot assays, and the corresponding cell culture medium is shown in Appendix A. All cells were kept in a humidified incubator at 37 °C and 5% CO_2_. Genomic DNA was extracted from all cell lines that were authenticated using STR screening to confirm its origin as thyroid cancer.

#### 4.1.2. Cell Treatments

##### EZH2 Pharmacological Inhibition

Ten thousand cells were seeded in 6-well plates and treated for 6 days with either EZH2 inhibitor EPZ6438 (Tazemetostat-Cayman Chemical) diluted in DMSO at 5.0 µM alone or in combination with MEK1/2 inhibitor U0126 (Promega, Madison, WI, USA) at 10 µM. EZH2 inhibitor was added to the medium every 2 days (6 days treatment) and on days 4 and 5, MAPK inhibitor was added (48 h total treatment).

### 4.2. EZH2 Gene Editing with CRISPR/Cas9

#### Plasmid Cloning

In order to permanently inhibit *EZH2* expression in ATC cell lines, we used the CRISPR/Cas9 system to induce DNA double-strand break (dsDNA) and nonhomologous end joining (NHEJ) in the human *EZH2* gene. For that, we designed three different single-guide RNA (sgRNA) that targeted flanking sequences at *EZH2*’s exon using the ChopChop tool (https://chopchop.cbu.uib.no/) [82]. sgRNAs sg7, sg18 and sg25 targeted, respectively, at 33 bp, 67 bp and 92 bp upstream of the start codon. These sgRNAs were cloned in pSpCas9(BB)-2A-Puro (PX459) plasmid, which has puromycin resistance, and named here as sg7, sg18 and sg25. Briefly, the annealed oligonucleotides (Appendix A) were ligated into linear plasmids digested with BbsI at a cloning site downstream of the U6 promoter, creating PX459-*EZH2*-sg7, PX459-*EZH2*-sg18 and PX459-*EZH2*-sg25. After confirming the correct insertion of the sgRNA sequence by Sanger sequencing, the plasmids were transfected in SW1736 and KTC2 cells using Lipofectamine^TM^ 2000 reagent (Invitrogen, Carlsbad, CA, USA). The cells were treated with 1 µg/mL puromycin for 7 days. Single-cell clones were isolated by limiting dilution plating. For that, we plated ~50 cells in a 10 cm dish and, after the colonies grew, we isolated about 9 individual colonies by trypsinization and then replated the cells into new plates to create SW1736-CRISPR-*EZH2* cell line clones (which we will herein name by SW1736-ClA, SW1736-ClC and SW1736-ClE). We expanded these clones for the in vitro and in vivo assays. The control group was generated by the transfection of empty PX459 with no sgRNAs cloned, creating SW1736-CTR cells.

For rescue assays, beta-catenin overexpression was induced with plasmid pCI-hβ-catenin [83]. As a control, cells were transfected with an empty pCDNA-3.1. Briefly, cells 5 × 10^4^ cells were seeded in 6 cm dishes and transfected with 1 µg of pCI-hβ-catenin or pCDNA-3.1 using Lipofectamine^TM^ 3000 reagent (Invitrogen). Five hours after transfection, cells were plated for colony formation assay.

### 4.3. Genomic DNA Sequencing for Gene Editing Validation

The genomic DNA was extracted from cells using DNeasy Blood & Tissue Kit (Qiagen, Hilden, Germany) and was used to PCR-amplify the genomic region targeted by CRISPR/Cas9 in the EZH2 gene. We also amplified the genomic segment of potential off-targets in *TMEM74B* and *MAD1L1* genes (Appendix A). The PCR products were sequenced using the Sanger method, and the analysis of gene editing was performed using the online tool SeqScreener Gene Edit Confirmation App (SGC—https://apps.thermofisher.com/apps/gea-web/#/setup (accessed on 27 May 2022)) by inputting the sequencing files (.ab format) from SW1736-CTR and edited cell clones.

### 4.4. Cell Function Assays

#### 4.4.1. Cell Count 

Growth curves were determined by seeding 2.5 × 10^4^ cells per well in 12-well plates and cultivating for 24, 48 and 72 h. After each of these periods, the cells were washed in PBS, detached by trypsinization, fixed in 3.7% formaldehyde and collected in new tubes. The average number of cells in the triplicate was determined by counting cells using a Guava Easycite Mini cytometer (Millipore, Burlington, MA, USA).

#### 4.4.2. Cell Invasion 

Invasion assay was performed using Transwell chambers with a membrane pore size of 8.0 μm (Corning Inc., Corning, NY, USA) precoated with 50 µL Geltrex TM Matrix (Gibco (Billings, MT, USA)). Fifty thousand SW1736-CTR and SW1736-ClA, ClC and ClE cells were suspended in a culture medium containing 0.5% FBS and plated in the upper chamber, whereas the lower chamber contained culture medium with 10% FBS as chemoattractant. After 48 h, noninvasive cells on the top chamber were removed using a cotton swab, and cells that invaded through the membrane were fixed in 3.7% formaldehyde diluted in PBS and stained with 1% crystal violet in 2% ethanol. Images were taken using a Nikon Eclipse E600 microscope equipped with an optical camera CF160 epifluorescence and the number of cells was counted at 100× magnification.

#### 4.4.3. Cell Migration

Migration assay was performed using uncoated Transwell chambers with a membrane pore size of 8.0 μm (Corning Inc.). Twenty-five thousand SW1736-CTR and SW1736-ClA, ClC and ClE cells were suspended in a culture medium containing 0.5% FBS and plated in the upper chamber, whereas the lower chamber contained culture medium with 10% FBS as chemoattractant. After 18 h, nonmigrating cells on the top chamber were removed using a cotton swab, and cells that migrated through the membrane were fixed in 3.7% formaldehyde diluted in PBS and stained with 1% crystal violet in 2% ethanol. Images were taken using a Nikon Eclipse E600 microscope equipped with an optical camera CF160 epifluorescence and the number of cells was counted at 200× magnification.

#### 4.4.4. Colony Formation 

Cells were seeded at very low density (300 cells) in 6-well plates and cultured for 8 days. After that period, the cells were fixed with formaldehyde 3.7% and stained with 1% crystal violet in 2% ethanol. Representative images were taken and colonies were counted using ImageJ.

### 4.5. Gene Expression Analysis

Total RNA was extracted using the phenol–chloroform method with the TRIzol reagent (Invitrogen-Thermo Fisher, Carlsbad, CA, USA). The reverse transcription of 1–3 μg of total RNA was performed using oligo-dT primer and MMLV reverse transcriptase (Invitrogen, Thermo Fisher Scientific, Waltham, MA, USA). The expression of the protein-coding gene was analyzed by quantitative PCR (qPCR) using SYBR Green Master Mix, cDNA, and specific primers (Appendix A) in a ViiA7^®^ Sequence Detection System (Applied Biosystems, Thermo Fisher Scientific, Waltham, MA, USA). Gene expression was normalized by comparison with *RPL19* levels and calculated using the QGene program [84] using the Ct data. 

For miRNA expression, 10 ng of total RNA was reverse transcribed using a TaqMan^®^ Reverse Transcription Kit (Applied Biosystems, Thermo Fisher Scientific, Waltham, MA, USA) in the presence of stem-loop primers; followed by qPCR using TaqMan MicroRNA Assays for *miR-200a* (assay 502), *miR-200c* (assay 2300) and RNU6B (assay 1093) (Applied Biosystems-Thermo Fisher Scientific, Waltham, MA, USA); and TaqMan Universal PCR Master Mix, No AmpErase^®^ UNG (Life Technologies, Thermo Fisher Scientific, Waltham, MA, USA) in a ViiA7 Sequence Detection System. MiRNA expression was normalized by comparison with *RNU6B* levels and calculated using the Qgene program.

### 4.6. Western Blot

Total protein was extracted from cells using RIPA buffer (20 mM Tris, pH 7.5, 150 mM NaCl, 1% Nonidet P-40, 0.5% sodium deoxycholate, 1 mM EDTA and 0.1% SDS) containing 10% protease inhibitor cocktail and 1% phosphatase inhibitor cocktail (Sigma, St. Louis, MO, USA). Protein concentration was determined using the Bradford assay (Bio-Rad Laboratories, Hercules, CA, USA), and 25 μg of each sample was fractionated by 10% SDS-PAGE and blotted onto a nitrocellulose Hybond-ECL membrane (Amersham Biosciences, Little Chalfont, UK). Nonspecific binding sites were blocked with 5% nonfat dry milk in Tris-buffered saline—0.1% Tween-20. The primary antibodies used are shown in Appendix A. Antibody bound to target protein was detected with horseradish-peroxidase-conjugated secondary antibodies and developed with luminol and p-coumaric acid (Sigma) reagents in the presence of hydrogen peroxide. Chemiluminescence emission was visualized with an ImageQuant LAS4000 imaging system (GE Healthcare, Little Chalfont, UK).

### 4.7. Luciferase Gene Reporter Assay

#### 4.7.1. Wnt-β-Catenin Signaling 

We used the reporter plasmid M50 Super 8× TOPFlash (pM50) containing seven copies of the sequence AGATCAAAGGgggta (the uppercase and lowercase correspond to a TCF/LEF-binding site and a spacer, respectively) upstream of the firefly luciferase gene. Briefly, 5 × 10^4^ cells were seeded into 24-well plates and transfected with 300 ng of pM50 plus 30 ng of pRL (*Renilla* luciferase) using Lipofectamine^TM^ 2000 (Invitrogen-Thermo Fisher). Cell lysates were collected after 24 h of transfection. Firefly and *Renilla* luciferase were measured using the Dual-Luciferase Reporter assay system (Promega) in GloMax^®^ 20/20 Luminometer (Promega). The plasmid M50 Super 8× TOPFlash was obtained from the Addgene repository: Addgene plasmid # 12456.

#### 4.7.2. miR-200 Promoter Reporter

Putative promoter regions for mouse cluster mir200c/141 or mir200b/a/429 were screened using the FANTOM5 program (https://fantom.gsc.riken.jp/5/ (accessed on 28 February 2023)) to identify the transcription start sites (TSS). The luciferase reporter plasmids were constructed by PCR cloning the region −1445–0 of mir200c/141 or −463–4027 of mir200b/a/429, and ligation into XhoI and BglII sites of pGl4.20 minP plasmid (Promega). The plasmids were sequenced to confirm the inserts. For the luciferase reporter assay, we used the same protocol as for the M50 plasmid described previously.

### 4.8. In Vivo Xenotransplant in Nude Mice

This study complied with the guidelines of the Institutional Animal Care and Use Committee (IACUC) of the Institute of Biomedical Sciences, University of Sao Paulo, registered as protocol number CEUA No. 2023150720.

For the injection in nude mice, 2 × 10^6^ SW1736-CTR and SW1736-ClE cells were resuspended in cold PBS and mixed 1:1 with Geltrex^TM^ Matrix (Corning, NY, USA) in a final volume of 100 µL that was injected into opposite flanks of nude mice (left: SW1736-CTR; right: SW1736-ClE). The evolution of tumor growth was accompanied for 35 days and tumor volume (V) was calculated using measurements of length (L) and width (W) of the tumor with calipers, using the formula V = (L × W^2^)/2. After euthanasia, the tumors were excised, weighed, fixed in buffer 3.7% formaldehyde and embedded in paraffin for histological and immunohistochemical analysis.

For IHC, 5 µm tumor slices were submitted to deparaffinization in xylene and hydration through a series of decreasing alcohol concentrations. The immunohistochemical procedure was performed by an indirect 3-stage immunoenzymatic method (Martins et al., 2002). Briefly, after endogenous peroxidase activity was blocked with 3% hydrogen peroxide for 15 min, tissues were washed in phosphate-buffered saline (PBS) and incubated with primary antibody overnight at 4 °C. The primary antibodies were diluted in Tris-buffered saline (TBS) and 0.05% bovine serum albumin (BSA). Then, slices were washed and incubated with biotinylated secondary antibody for 2 h at room temperature. After washing, slices were incubated with streptavidin–peroxidase for 2 h at room temperature. The reaction was revealed by a mixture of 3,3′-diaminobenzidine with hydrogen peroxide. The sections were then counterstained with Gill’s hematoxylin. The immunopositivity of the reaction was detected as brown staining observed by light microscopy. We photographed the cells using a Nikon Eclipse E600 microscope equipped with an optical camera CF160 epifluorescence and total and positive cells were counted for anti-Ki67 immunostaining and relative area for positive anti-αSMA cells at 200× magnification.

### 4.9. Analysis of Publicly Available Data from Thyroid Tumors 

#### 4.9.1. The Cancer Genome Atlas (TCGA) Data

For mRNA (RNASeqV2) gene expression of PRC2 genes (*EZH2*, *EED*, *SUZ12*) in PTC was extracted from the TCGA web portal (accessed at https://www.cbioportal.org/) from 59 paired-matched nontumoral and PTC samples as previously described [85].

#### 4.9.2. Microarray Data

The National Center for Biotechnology Information Gene Expression Omnibus (GEO) provided access to gene expression microarray data from Affymetrix Human Genome U133 Plus 2.0 platform. The selection of studies was performed by using the search terms “thyroid cancer” and “array”. Fresh and frozen primary samples were considered as criteria for selection. Studies from thyroid cancer cell lines, nonhuman thyroid models and radiation were excluded. The raw files (.Cel) of 31 anaplastic and normal human thyroid samples were imported from GEO (GSE76039, GSE60542, GSE33630 and GSE3678), and probe-level summarization files (.CHP) were obtained by using Robust Multi-array Analysis (RMA) in Expression Console Software (Affymetrix). The outlier samples were identified according to quality control metrics and removed. Differentially expressed genes list was obtained as fold change (ATC vs. NT) from Transcriptome Analysis Console (TAC) software (Affymetrix). 

#### 4.9.3. Cistrome Data Browser 

Interaction of both EZH2 and H3K27me3 with the CpG islands from thyroid-differentiation genes was analyzed at the Cistrome database for ChIP-seq and DNA-seq datasets (http://cistrome.org/db/). The following CistromeDB ID studies were selected using EZH2 as a factor: 41,788; 41,794; 8664; 74,684; 9009. The following CistromeDB ID studies using H3K27me3 as a factor: 45,141; 71,328; 71,329; 52389; 52,394.

### 4.10. Statistical Analysis

The results were presented as the mean ± standard deviation (SD) and were submitted to analysis of variance followed by a t-test or the Tukey test. Differences were considered significant at *p* ≤ 0.05.

## 5. Conclusions

We show that anaplastic thyroid carcinoma exhibits an elevated expression of EZH2, and its inhibition through either CRISPR/Cas9-induced *EZH2* gene editing or with EPZ6438 inhibitor improves thyroid follicular-cell differentiation and induces mesenchymal–epithelial transition. Furthermore, we have observed a potent antitumoral effect of EZH2 inhibition, as evidenced by reduced cell count, migration and invasion in vitro, and inhibited tumor growth in a xenograft mouse model. Taken together, our findings suggest that targeting EZH2 may be an effective adjuvant therapy for ATC, given its role in regulating cell differentiation and EMT, which are critical processes in ATC pathogenesis.

## Figures and Tables

**Figure 1 ijms-24-07872-f001:**
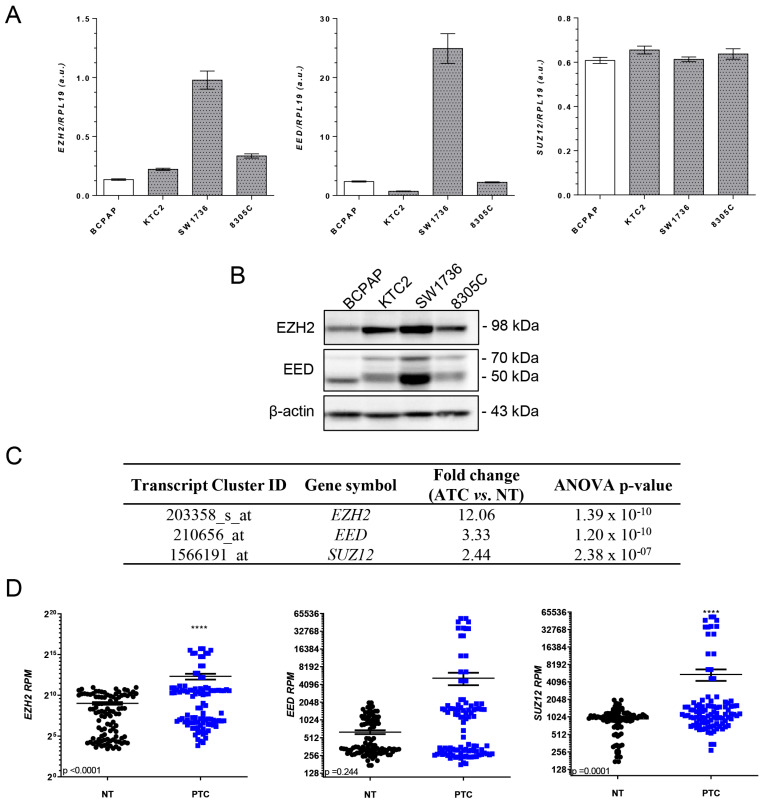
EZH2/PRC2 proteins are overexpressed in ATC cells. (**A**) qPCR analysis of *EZH2*, *EED* and *SUZ12* mRNA levels in a panel of thyroid cancer cells. The expression was normalized using *RPL19*. White bars represent the PTC cell line BCPAP, and gray bars represent the ATC cell lines KTC2, SW1736 and 8305C. (**B**) Western blot analysis of EZH2 and EED expression in a panel of thyroid cancer cells. The expression was normalized using β-actin. (**C**) Gene expression of PRC2 genes in ATC patients compared to normal thyroid (NT) in microarray data (accessed at Gene Expression Omnibus—GEO: GSE76039, GSE60542, GSE33630 and GSE3678). (**D**) Gene expression of PRC2 genes (*EZH2*, *EED* and *SUZ12*) in PTC extracted from the TCGA database from 59 paired-matched nontumoral and PTC samples (accessed https://www.cbioportal.org/). Data expressed as reads per million (RPM). a.u., arbitrary unit. ****, *p* < 0.0001.

**Figure 2 ijms-24-07872-f002:**
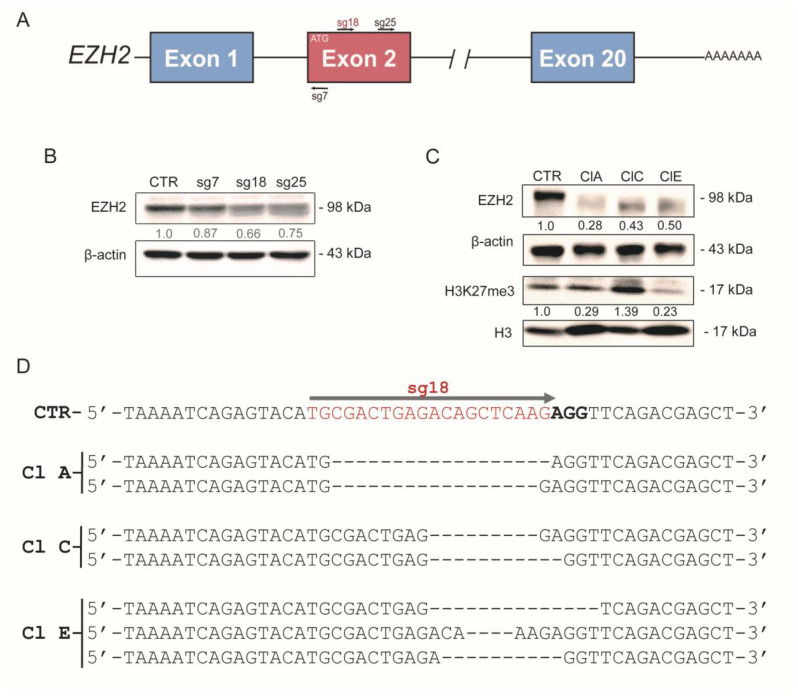
Targeting the *EZH2* gene with CRISPR/Cas9. (**A**) Graphical representation of *EZH2* gene with introns and exons. At the second exon, near the *ATG* start codon, the sgRNAs (sg7, sg18 and sg25) targeted the *EZH2* gene at different positions. (**B**) Western blot validation of EZH2 protein levels in SW1736 mixed populations after transfection with sgRNA (sg)-7, 18 and 25. (**C**) Protein levels of EZH2 and H3K27me3 in clonal populations (Cl) ClA, ClC and ClE derived from CRISPR/Cas9-edited SW1736-sg18 cells. EZH2 expression was normalized using β-actin; H3K27me3 was normalized by total H3 expression. (**D**) Genomic DNA sequencing of SW1736-sg18 clones ClA, ClC and ClE after CRISPR/Cas9-mediated *EZH2* gene editing. sgRNA-guided target regions are highlighted in red and the PAM sequence in bold as a graphical example in SW1736-CTR. The dsDNA repair resulted in 18 nt and 17 nt deletion in clone A (ClA), 9 nt and 11 nt deletion changes in clone C (ClC) and 14 nt, 4 nt and 10 nt deletion in clone E (ClE) *EZH2* gene. Sg, single guide; CTR, control; cl, clone; PAM, protospacer adjacent motif; dsDNA, double-strand break DNA.

**Figure 3 ijms-24-07872-f003:**
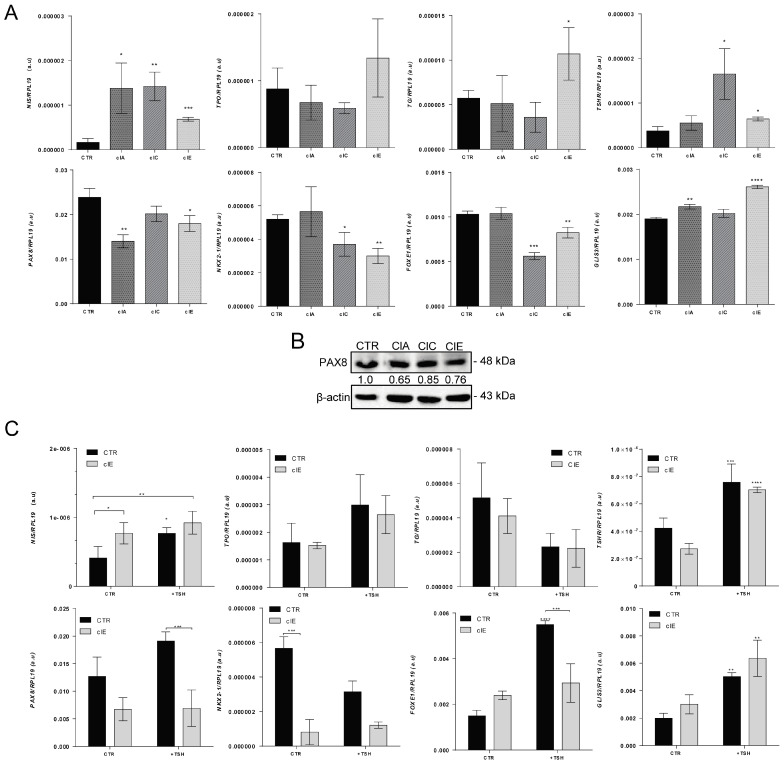
CRISPR/Cas9-mediated *EZH2* gene editing improves thyroid differentiation of ATC cells. (**A**) RT-qPCR analysis of thyroid-differentiation genes’ expression *NIS*, *TPO*, TG and *TSHR*, and transcription factors *PAX8*, *NKX2-1*, *FOXE1* and *GLIS3* in SW1736-ClA, SW1736-ClC and SW1736-ClE compared with SW1736-CTR. (**B**) Western blot analysis of PAX8 expression in SW1736-edited cells. (**C**) RT-qPCR analysis of gene expression of thyroid-differentiation genes response to exogenous TSH treatment (3 mU/mL) in SW1736-CTR and SW1736-ClE. The expression was normalized using *RPL19*. Data are expressed as mean  ±  SD (*n*  =  3) for gene expression. a.u., arbitrary units; *, *p* < 0.05; **, *p* < 0.01; ***, *p* < 0.001; ****, *p* < 0.0001 vs. SW1736-CTR or nontreated SW1736-CTR and SW1736-ClE.

**Figure 4 ijms-24-07872-f004:**
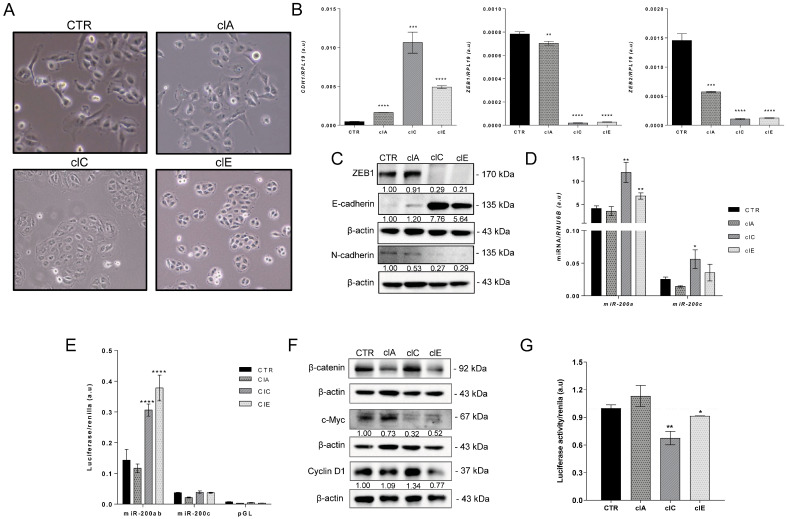
CRISPR/Cas9-mediated EZH2 gene editing induces mesenchymal–epithelial transition (MET) in ATC cells. (**A**) Phase contrast images of SW1736-CTR and SW1736-ClA, ClC and ClE cells in culture showing distinct morphology. SW1736-CTR and SW176-ClA present mesenchymal-like phenotype, whereas SW1736-ClC and SW1736-ClE are more epithelial-like. Images were taken at 100× magnification. (**B**) Gene expression of epithelial gene *CDH1* (E-cadherin) and mesenchymal transcription factors *ZEB1* and *ZEB2* in SW1736-ClA, SW1736-ClC and SW1736-ClE compared with SW1736-CTR. The expression was normalized using *RPL19*. Data are expressed as mean  ±  SD (*n*  =  3) for gene expression. (**C**) Protein levels of ZEB1, E-cadherin and N-Cadherin by Western blot in the whole-cell lysate of SW1736-ClA, SW1736-ClC and SW1736-ClE compared to SW1736-CTR. (**D**) RT-qPCR gene expression of mature *miR-200a* (**left**) and *miR-200c* (**right**) in SW1736-ClA, SW1736-ClC and SW1736-ClE compared with SW1736-CTR. *RNU6B* was used as endogenous control. Data are expressed as mean  ±  SD for gene expression. (**E**) Activation of *miR-200* promoter reporter luciferase in SW1736-edited cells compared to SW1736-CTR. (**F**) Protein levels of β-catenin, c-MYC and Cyclin D1 by Western blot in the whole-cell lysate of SW1736-ClA, SW1736-ClC and SW1736-ClE compared to SW1736-CTR. (**G**) Activation of Wnt/β-catenin signaling in SW1736-edited cells compared to SW1736-CTR using pM50 luciferase reporter that contains TCF/LEF binding sites. Luciferase activity was normalized using renilla activity. Data are expressed as mean  ±  SD (*n*  =  4). a.u., arbitrary units; *, *p* < 0.05; **, *p* < 0.01; ***, *p* < 0.001; ****, *p* < 0.0001 vs. SW1736-CTR.

**Figure 5 ijms-24-07872-f005:**
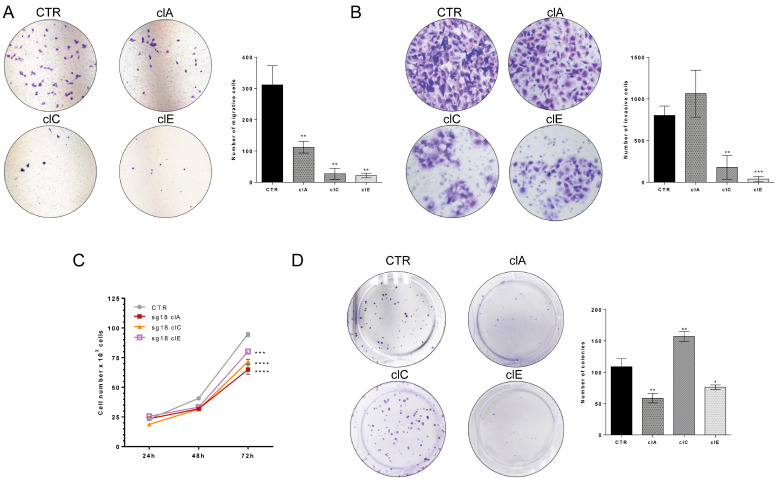
CRISPR/Cas9-mediated EZH2 gene editing inhibits ATC cells’ growth, migration and invasion in vitro. (**A**) Cell migration assay in Transwell^®^ chambers in SW1736-CTR cells compared to SW1736-ClA, ClC and ClE. Representative images of migratory cells. The graph shows quantification of migratory cells in four random areas at 100× magnification. (**B**) Cell invasion assay using Transwell^®^ chambers covered with GELTREX^®^ in SW1736-CTR cells compared to SW1736-ClA, ClC and ClE. The graph shows quantification of invasive cells in four random areas at 200× magnification. (**C**) Cell counting of SW1736-ClA, ClC and ClE. Cell number was accessed after 24 h, 48 h and 72 h in culture compared to SW1736-CTR. (**D**) Colony formation assay of SW1736-CTR compared to SW1736-ClA, ClC and ClE. The graph shows quantification of the number of colonies. The results are representative of two independent experiments performed in triplicate. *, *p* < 0.05; **, *p* < 0.01; ***, *p* < 0.001; ****, *p* < 0.0001 vs. SW1736-CTR.

**Figure 6 ijms-24-07872-f006:**
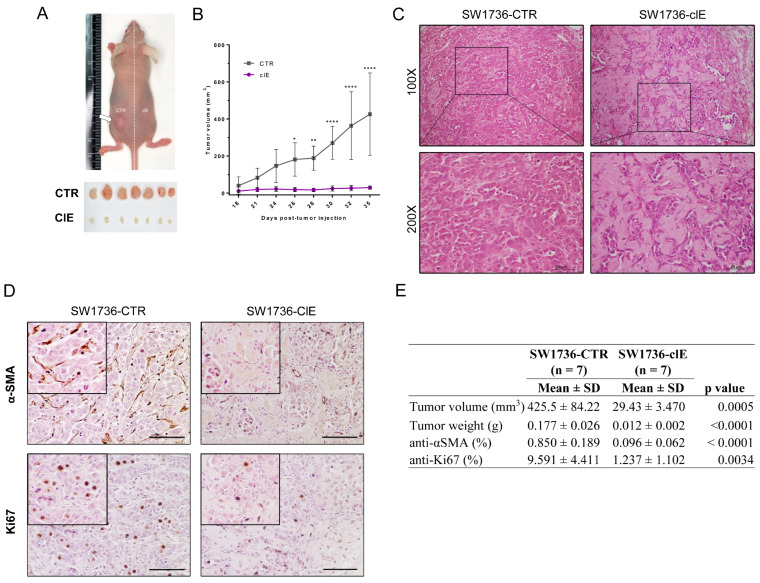
CRISPR/Cas9-mediated EZH2 gene editing inhibits ATC tumor growth in vivo. (**A**) Representative image of an animal injected with SW1736-CTR cells (in the left flank) and SW1736-ClE cells (in the right flank), and alignment of tumors collected from seven animals (n = 7) after euthanasia and surgical excision (each animal in a column). (**B**) Tumor volume evolution from day 18 (first visible tumor) to the euthanasia endpoint at day 35. (**C**) Histology of SW1736-CTR tumor (**left**) and SW1736-ClE tumor (**right**) stained with H&E. Magnification: 100× and 200×. (**D**) Immunohistochemical analysis of anti-αSMA (**upper** panel) and anti-Ki67 (**lower** panel) from section of SW1736-CTR (**left**) and SW1736-ClE (**right**) tumors counterstained with Gill’s hematoxylin at 200× magnification. Scale bar: 60 um. (**E**) Quantification of tumor parameters of volume and weight at the endpoint, and quantification of IHC staining for Ki67 and αSMA: Total and positive cells were counted for anti-Ki67 immunostaining and relative area for positive anti-αSMA area. *, *p* < 0.05; **, *p* < 0.01; ****, *p* < 0.0001 vs. SW1736-CTR.

**Figure 7 ijms-24-07872-f007:**
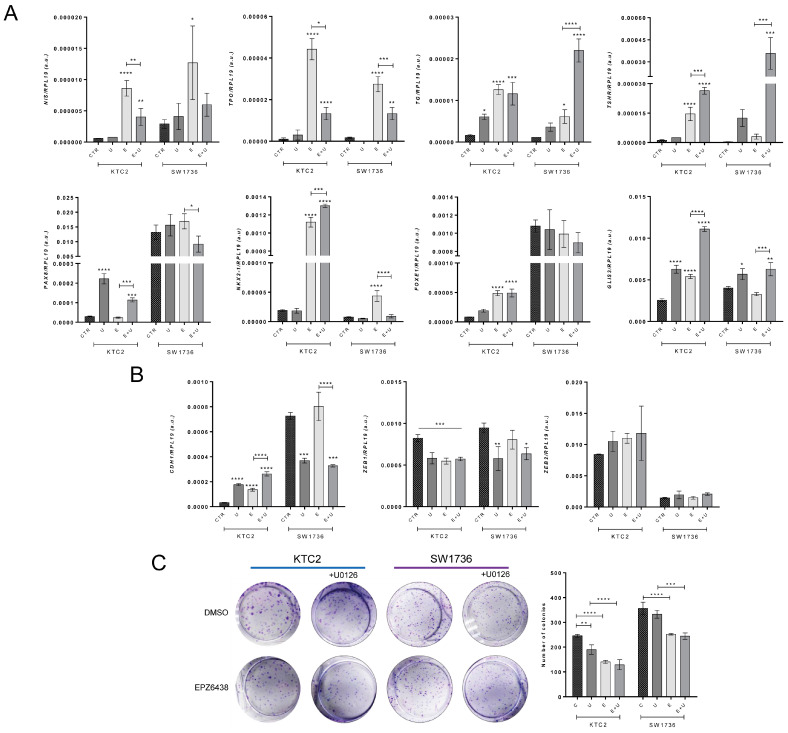
Pharmacological blockage of EZH2 activity induces differentiation of ATC cells and reduces colony formation. KTC2 and SW1736 cells were treated with 5.0 µM of EPZ6438 alone (E) or in combination (E + U) with 10 µM of U0126 (U) compared with DMSO control (CTR). (**A**) Gene expression of thyroid-differentiation genes *NIS*, *TPO*, *TG* and *TSHR*; transcription factors *PAX8*, *NKX2-1*, *FOXE1* and *GLIS3*. (**B**) Gene expression of epithelial gene *CDH1* and mesenchymal genes *ZEB1* and *ZEB2*. Data are expressed as mean  ±  SD (*n*  =  3) for gene expression. (**C**) Colony formation assay of KTC2 and SW1736. Data are expressed as mean ± SD (*n*  =  3) for total colony count. a.u., arbitrary units; *, *p* < 0.05; **, *p* < 0.01; ***, *p* < 0.001; ****, *p* < 0.0001 vs. DMSO.

**Figure 8 ijms-24-07872-f008:**
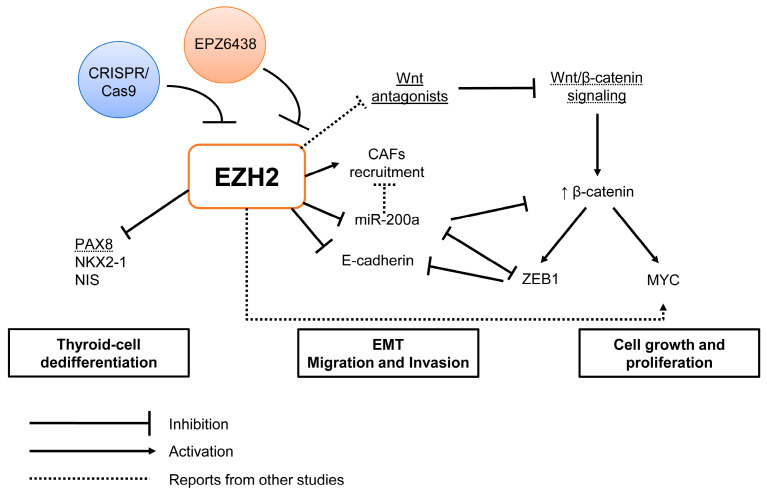
Graphical summary of the study. EZH2 may act in different steps of anaplastic thyroid cancer aggressiveness, such as dedifferentiation, cell migration, invasion and proliferation. In thyroid cell dedifferentiation, EZH2 by H3K27me3 silences PAX8, NKX2-1 and NIS genes. In EMT signaling, EZH2 inhibits E-cadherin and *miR-200a* expression and induces ZEB1 and ZEB2 mesenchymal genes to induce EMT. Moreover, downregulation of *miR-200a* induces the recruitment of cancer-associated fibroblasts (CAFs), which are associated with poor outcome and metastasis. Finally, EZH2 modulates Wnt/β-catenin signaling, which, besides EMT, is associated with cancer cell growth and proliferation. Here, we show that inhibition of EZH2 expression and/or its methyltransferase activity with CRISPR/Cas9 or EPZ6438, respectively, improves thyroid cell differentiation, induces MET and inhibits cell proliferation.

## Data Availability

The data presented in this study are available in the current article and data mining is described in the Section 4.

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
