# Peer review of "Modulation of EZH2 Activity Induces an Antitumoral Effect and Cell Redifferentiation in Anaplastic Thyroid Cancer"

_ijms, 2023, doi:10.3390/ijms24097872_

Round 1

Reviewer 1 Report

The authors used CRISPR/CAS9 method to modify the levels of EZH2 and H3K27me3 in human thyroid cancer cell lines and evaluated the effect of EZH2 downregulation on cancer cells survival bot in vitro and in vivo and tried to explain these results with various cellular and molecular methods. Some issues need to be addressed before the submission acceptance.

1.      Why the CRISPR/CAS9 experiments with KTC2 cell line were not continued, despite that the authors already prepared KTC2 clones with knocked out EZH2? What are the genetic and phenotypic differences between KTC2 and SW1736 thyroid cancer cell lines? It would be great to run paralelly all / most of the experiments depicted on figures 3 – 6 in two different thyroid cancer cell lines to confirm the results more "globally"

2.      The SW1796 engineered clones behave very diverse, depending on EZH2 and HeK27me3 levels, which makes the results suspicious and difficult to interpret. For instance, why ClA clone behaves in most experiments exactly like ctrl cells, although the EZH2 expression in ClA clone is drastically diminished?  Also, despite diminished levels of EZH2 in ClC clone, the H3K27me3 expression in this clone is even higher than in ctrl cells. However, most of the experiments show similar results as for ClE clone, where both EZH2 and H3K27m3 is drastically diminished. On Figure 5D ClC cells behave like ctrl cells, although different EZH2 levels, but quite similar levels of H3K27me3 (still refering to figure 2C). Can the authors explain such variable results? Is it possible, that other histone methylases complement lack of EZH2 in PRC2 histone methylating complex? For instance EZH1? Did the authors try to rule out the involvement of EZH1 in PRC2-mediated H3K27 methylation? Also, it is possible, that the expression of chosen examined genes is regulated by other mechanisms? The authors should also discuss PRC2- and histone methylation- independent roles of EZH2 in carcinogenesis (as outlined for EZH2-c-Myc interaction). As for the results, the authors should focus only on those clones where EZH2 and H3K27me3 diminution is evident and compare them with ctrl cell lines (as it has been done on Figure 6). Such experiments should also contain KTC2-control cells and KTC2 cell lines with silenced EZH2 and diminished H3K27me3 levels (as mentioned above).

3.      Referring to this part: “The effectiveness of CRISPR/Cas9-mediated gene editing may be influenced by factors such as gene copy number variation, in addition to sgRNA+Cas9 efficiency and input [58]. Indeed, SW1736 cells are tetraploid and have amplification in the 7q arm of chromosome 7 where EZH2 is located, indicating the possibility of additional copies of EZH2-gene [36]”. Again: then why these results with Cas9 methods were not compared to KTC2 cell lines?

Some minor points:

-          most of bar plots are difficult to read due to font type and size. Modify bar plot labels throughout the manuscript

-          url or reference for this tool FANTOM5 (materials and methods)

-          Data Availability Statement and Supplementary Material are not adequately modified from MDPI template (Page 20)

Reviewer 2 Report

The authors set out to determine the role and functional significance of EZH2 in anaplastic thyroid cancer (ATC). They found that ATC cell lines and patient tumors express higher levels of EZH2 compared to papillary thyroid (PTC) and normal tissues respectively. Using CRISPR/Cas9-mediated knockout of EZH2, they reported that EZH2 knockout cells have reduced migration and invasion, and reduced growth in vitro and in immunocompromised mice. Finally, they validated some of the findings using EZH2 inhibitor EPZ6438.

Overall, the findings are new, straightforward, and easy to follow. However, a few major flaws are apparent, and need to be addressed before further consideration:

1. Successful knockout of EZH2 should result in reduction of H3K27me3. The data in Figure 2C do not seem supportive of this, despite the provided quantification. Can the authors perform some chIP experiment to quantify the levels of H3K27me3?

2. The rationale of combining EZH2 inhibitor and MEK inhibitor is not clear.

3. It is conceivable, based on in vitro data, that the CRISPR clones will grow slower in vivo and hence less Ki67 positive. But why lower number of aSMA? Are these aSMA+ cells proliferating less? Do the CRISPR tumor cells secrete less cytokines, hence less infiltrating fibroblasts?

2. Overall, figure quality is relatively poor, resulting in difficulty in interpreting the data. For example, most labeling for qRT-PCR data is almost illegible.

3. Minor: figure 3: why are there no error bars for C1A on TGFb graph? No error bars for C1A and C1C on TSHR graph? It is also not convincing, based on the western blot provided, that PAX8 goes down in the CRISPR clones compared to the control.

4. Minor: Fig. 6- double check on scale bar? Indicated as 20 µm. A typical nucleus (that stains positive for Ki67) should be around ~10 µm.

Round 2

Reviewer 2 Report

The authors have addressed all of my concerns.